# Comparison between Three Different Techniques for the Detection of EGFR Mutations in Liquid Biopsies of Patients with Advanced Stage Lung Adenocarcinoma

**DOI:** 10.3390/ijms24076410

**Published:** 2023-03-29

**Authors:** Milena Casula, Marina Pisano, Panagiotis Paliogiannis, Maria Colombino, Maria Cristina Sini, Angelo Zinellu, Davide Santeufemia, Antonella Manca, Stefania Casula, Silvia Tore, Renato Lobrano, Antonio Cossu, Giuseppe Palmieri

**Affiliations:** 1Unit of Cancer Genetics, Institute of Genetic Biomedical Research (IRGB), National Research Council (CNR), 07100 Sassari, Italy; casulam@yahoo.it (M.C.) marina.pisano@cnr.it (M.P.); maria.colombino@cnr.it (M.C.); mariacristina.sini@cnr.it (M.C.S.); antonella.manca@cnr.it (A.M.); stefania.casula@cnr.it (S.C.); silvia.tore@cnr.it (S.T.); 2Anatomic Pathology and Histology, University Hospital (AOU) of Sassari, 07100 Sassari, Italy; ppaliogiannis@uniss.it (P.P.); renato.lobrano@gmail.com (R.L.); cossu@uniss.it (A.C.); 3Department of Biomedical Sciences (DSB), University of Sassari, 07100 Sassari, Italy; azinellu@uniss.it; 4Medical Oncology, Civil Hospital, 07041 Alghero, Italy; davidesanteufemia@gmail.com; 5Immuno-Oncology & Targeted Cancer Biotherapies, University of Sassari, 07100 Sassari, Italy

**Keywords:** lung adenocarcinoma, EGFR, mutation analysis, tyrosine kinase inhibitors

## Abstract

Oncogenic mutations in the *EGFR* gene are targets of tyrosine kinase inhibitors (TKIs) in lung adenocarcinoma (LC) patients, and their search is mandatory to make decisions on treatment strategies. Liquid biopsy of circulating tumour DNA (ctDNA) is increasingly used to detect EGFR mutations, including main activating alterations (exon 19 deletions and exon 21 L858R mutation) and T790M mutation, which is the most common mechanism of acquired resistance to first- and second-generation TKIs. In this study, we prospectively compared three different techniques for *EGFR* mutation detection in liquid biopsies of such patients. Fifty-four ctDNA samples from 48 consecutive advanced LC patients treated with TKIs were tested for relevant *EGFR* mutations with Therascreen^®^ EGFR Plasma RGQ-PCR Kit (Qiagen). Samples were subsequently tested with two different technologies, with the aim to compare the EGFR detection rates: real-time PCR based Idylla™ ctEGFR mutation assay (Biocartis) and next-generation sequencing (NGS) system with Ion AmpliSeq Cancer Hotspot panel (ThermoFisher). A high concordance rate for main druggable *EGFR* alterations was observed with the two real-time PCR-based assays, ranging from 100% for T790M mutation to 94% for L858R variant and 85% for exon 19 deletions. Conversely, lower concordance rates were found between real-time PCR approaches and the NGS method (L858R: 88%; exon19-dels: 74%; T790M: 37.5%). Our results evidenced an equivalent detection ability between PCR-based techniques for circulating *EGFR* mutations. The NGS assay allowed detection of a wider range of *EGFR* mutations but showed a poor ability to detect T790M.

## 1. Introduction

Lung cancer is one of the most common malignancies and a leading cause of cancer mortality; more than 2 million new cases per year are estimated worldwide, accounting for 11.4% of all cancer types, along with approximately 1,800,000 deaths, which represent 18% of all cancer deaths estimated worldwide [1]. Non-small cell lung cancer (NSCLC) comprises approximately 85–90% of all lung cancer types and in the last decade has been widely investigated in the development of modern targeted therapies based on the identification of genetic alterations that are responsible for the oncogenic process [2,3]. The epidermal growth factor receptor (*EGFR*) gene has been found mutated in about 15% of Caucasian patients, and up to 50% or more in Asian patients with lung adenocarcinomas [4], and its driver mutations were the first ones to be targeted in lung cancer.

The *EGFR* gene is a trans-membrane glycoprotein, member of the epidermal growth factor tyrosine kinase (TK) receptor family, and has been identified as a driving gene of NSCLC. Specific ligands binding to EGFR cause protein dimerization and tyrosine auto-phosphorylation, which activate a downstream cascade of pathways triggering cellular proliferation and survival [5]. The human *EGFR* gene is structured in 28 exons, with exons 18 to 21 encoding for the tyrosine kinase domain; indeed, most of the activating “hot spot” mutations involve sequence modifications in these exons [6]. Cancer cells harbouring deletions in exon 19 or point mutations in exon 21 (L858R, L861Q and others) and exon 18 (G719X and others) are associated with sensitivity to EGFR tyrosine kinase inhibitors (TKIs) [7,8,9]. Exon 19 deletions and L858R point mutation in exon 21 account for approximately 90% of EGFR activating mutations and represent the most common druggable genetic alterations in NSLCL patients [10,11].

The acquired drug resistance against first- and second-generation EGFR TKIs is mostly heterogeneous and based on multiple causative mechanisms: changes in EGFR mutational status, activation of alternative bypass or downstream pathways, and changes in the phenotype [12]. Among others, the acquired T790M mutation in exon 20 represents the most common mechanism of resistance to targeted therapies [12,13,14]. T790M leads to steric modifications which block binding of these medications at the kinase domain, leading to acquired resistance and disease progression [15]. Recently, the development of a third-generation anti-EGFR TKI (Osimertinib) showed excellent results in treatment of T790M-positive patients with lung adenocarcinoma [16,17]. Surprisingly, the persistence of T790M mutation in patients treated with Osimertinib has been shown to be a positive prognostic factor [18]. Therefore, T790M remains a valid prognostic biomarker in both patients treated with older or last-generation anti-EGFR TKIs. In addition to the intrinsic drug resistance, additional acquired mutations—such as EGFR-C797S or EGFR L718Q—have been found to drive acquired resistance to Osimertinib [19,20,21,22], promoting the development of novel fourth-generation EGFR inhibitors.

Tumour biopsy is the preferred approach for *EGFR* molecular testing, but in up to 30% of the cases, tissues are not available or their quality is inadequate for molecular testing [23,24]. Analysis of circulating tumour DNA (ctDNA) in liquid biopsy specimens provides a non-invasive surrogate for the detection of somatic mutations and is currently recommended by clinical guidelines for the monitoring of NSCLC targeted therapies [25]. ctDNA testing reduces the frequency of false positive results due to fixation-induced degradation and deamination of formalin-fixed and paraffin-embedded (FFPE)-derived nucleotide acids from tissue samples; in addition, ctDNA reflects the global tumour activity, and is not limited by the neoplastic heterogeneity which generally limits tissue biopsies. On the other hand, the amount and purity of tumour nucleic acids in the blood stream are often low, and this gives rise to several technical issues regarding their determination. A number of studies investigated the diagnostic accuracy of ctDNA in patients who progressed after anti-EGFR TKIs, and showed a very wide range of concordance with tumour tissue analyses [26,27].

Several technologies have been developed to carry out *EGFR* molecular testing. The Therascreen^®^ EGFR RGQ PCR kit is a real-time polymerase chain reaction test kit for the qualitative detection of exon 19 deletions and L858R mutations of the human EGFR gene [28] which has been approved by EU IVD Directive 98/79/EC and is widely used in clinical practice. Other PCR-based and, recently, next-generation sequencing (NGS) platforms also have been developed. In this study, we analysed 54 consecutive blood samples from 48 patients with advanced lung adenocarcinoma, carrying a druggable EGFR mutation, at progression time after treatment with first- or second-generation EGFR TKIs, using the Therascreen^®^ EGFR RGQ PCR kit and compared the results with those obtained using two additional screening techniques: the Biocartis Idylla™ test and an NGS-based assay. The aim was to investigate the diagnostic performances and concordance rates between these routinely used EGFR testing methods by also investigating their mutual ability to detect the circulating T790M variant as acquired drug resistance mutation.

## 2. Results

### 2.1. Patient Cohort

The main demographic and clinical features of the 48 patients enrolled are summarized in Table 1. All patients were affected by advanced stage adenocarcinoma (surgically unresectable stage III or metastatic stage IV) harbouring a druggable *EGFR* mutation; most patients had a single metastatic site involved (29/48; 60%) and underwent treatment with a first- (Gefitinib) or second- (Afatinib) generation anti-EGFR TKI (Table 1).

### 2.2. Mutation Analysis

The Therascreen^®^ EGFR Plasma kit, which is the CE-IVD approved method for *EGFR* mutational status testing in laboratory practice, was used in all cases from the series (N = 48) to establish the molecular diagnosis, as recommended by national clinical guidelines. For some patients, more than one ctDNA specimen was analysed due to the clinical need—i.e., sequential disease progression—requiring multiple T790M determinations before treatment decision. Overall, 54 ctDNA samples from 48 patients underwent mutation screening. Of note, the Therascreen^®^ EGFR Plasma kit is made to detect only circulating main druggable alterations (exon 19 deletions and exon 21 L858R mutation) as well as the T790M variant associated with TKI resistance. Figure 1 depicts the results of testing with Therascreen^®^ in blood samples from the series at the time of disease progression, reporting the subtypes of EGFR mutations detected by this assay.

Table 2 reports the comparison between the results obtained in blood samples with this method and the results of *EGFR* mutational testing by the pyrosequencing approach performed before treatment initiation in tissue samples. No statistically significant differences were found in exon 19 deletions and exon 21 L858R mutations, even when the patients were divided in accordance with the number of metastatic sites (one or more) or the stage of the disease (IIIB or IV) (Table 3). Of note, the presence of the somatic driver oncogenic mutation in ctDNA samples at the time of disease progression was mostly prevalent according to the tumour load: 13/17 (76.5%) in cases with ≥2 metastatic sites vs. 14/26 (53.8%) in those with 1 metastatic site; 16/24 (66.7%) in stage IV cases vs. 11/19 (57.9%) in stage IIIB cases (Table 3).

Concordance rates between the three screening methods on ctDNA samples are reported in Table 4.

Results obtained with the Therascreen^®^ assay were compared to those determined with Idylla™ and NGS approaches. Overall, 49/54 tests (90.7%) were concordant between Therascreen^®^ and Idylla™, while concordance with NGS assay was 74.1% (40/54 tests), a difference that was statistically significant (*p* < 0.001). In addition, Idylla™ presented the highest agreement percentage with Therascreen^®^ in mutated cases (23/27; 85.2%), compared with the NGS approach (13/27; 48.1%, *p* < 0.001). In non-mutated cases, rates were quite identical for both techniques compared with Therascreen^®^ (Table 4).

The *EGFR* sequence variations predictive of responsiveness or resistance to TKIs that were determined with Therascreen^®^ were subsequently compared with those identified with Idylla™ and the NGS-based assays (Table 5).

Total agreement was observed for the T790M mutation, while high concordance was found for both exon 19 deletions and L858R missense mutation in exon 21 (85.2% and 94.4%, respectively). We also focused on these genetic alterations to compare the NGS-based analysis with Therascreen^®^. Our results revealed a poorer level of detection of NGS for both the T790M variant (8/54 vs. 4/54, agreement rate 37.5%) and the exon 19 deletions (27/54 vs. 20/54, agreement rate 74.1%) in comparison with Therascreen^®^; nevertheless, the differences in the detection rates were not statistically significant. Appendix A provides detailed information on the concordance between the techniques under investigation in detecting each single genetic alteration. Globally, the results of 38/54 (70.4%) tests were in complete agreement, because the state of the genetic alterations searched was confirmed with all of the techniques used (12 mutated cases and 26 without any mutation). Contrarily, 16/54 samples (29.6%) were discordant for at least one of the variants analysed.

## 3. Discussion

First-generation TKIs against *EGFR* mutated NSCLC entered development in the 1990s and entered clinical practice in 2015 when the Food and Drug Administration (FDA) approved Gefitinib on the basis of the results of several trials which showed improved outcomes in comparison to traditional chemotherapy [29,30]. Initial enthusiasm for the results of first-generation TKIs soon waned, because it was seen that nearly all patients would eventually progress. The IMPRESS study showed that resistance and disease progression were associated with a specific secondary mutation in the *EGFR* gene, T790M (HR 1.49, *p* = 0.043 for T790M+ patients vs. HR 1.15, *p* = 0.609 in T790M− patients) [31]. Moreover, the first-generation TKIs including Gefitinib and Erlotinib, which are reversible inhibitors, were immediately joined by second-generation TKIs such as Afatinib, which acts as an irreversible, potent, and selective inhibitor of the ErbB tyrosine kinase family [32]. Thereafter, routine determination of T790M in liquid biopsies of patients treated with first- and second-generation TKIs entered progressively into clinical practice for early identification of the mutation-dependent inhibitor resistance; indeed, re-biopsy studies showed that T790M is detected in about two-thirds of progressed patients [33,34]. In addition, a recent study showed a higher incidence of the T790M mutation in patients who progressed under Gefitinib/Erlotinib, in comparison to those treated with Afatinib, suggesting that T790M incidence differs among first- and second-generation TKIs [35].

The advent of the third-generation TKI Osimertinib, which is an additional irreversible EGFR inhibitor of both all sensitizing mutations and the TKI resistance mutation T790M, was the most recent step in the targeted therapy of NSCLC. As mentioned before, the maintenance of T790M in liquid biopsies of patients treated with Osimertinib was shown to be a positive prognostic factor [18], despite some contrasting results suggesting that patients in whom an *EGFR* T790M mutation was detected in plasma samples demonstrated a poorer response to Osimertinib than those in whom the mutation was detected in tissue specimens [36]. In addition, Lin et al. showed that the co-occurrence of a T790M mutation with some rare *EGFR* mutations has a negative impact on the prognosis of patients with advanced adenocarcinoma [37].

In our previous experience, the survival curve of Sardinian lung adenocarcinoma patients carrying even very low rates (≤1.5%) of EGFR-T790M variants at baseline was significantly poorer in comparison to that of cases without such a mutation [38].

EGFR mutations can be determined in both tissue and liquid biopsies. Tumour tissue remains the largely preferred choice for the identification of molecular targets at the time of treatment decision (at baseline), whereas liquid biopsy through circulating free DNA (cfDNA) analysis represents an alternative option when tissue is unavailable or inadequate or the patient has comorbidities that exclude an invasive diagnostic approach. At disease progression, in patients undergoing treatment with target drugs—such as the anti-EGFR TKIs, mutation analysis based on cfDNA may even provide the advantage of better representing tissue heterogeneity toward the identification of mechanisms of acquired resistance, as recently stated by expert panel recommendations [39].

Therefore, novel technologies are continuously being introduced into clinical practice in order to address the technical issues bound to the small amounts of ctDNA and low levels of mutated alleles. The technologies for the application of the minimally invasive liquid biopsy actually include digital droplet PCR (ddPCR) systems, amplification refractory mutation system-PCR (ARMS-PCR), and others, which allow the detection of somatic mutations even with low variant allele fraction (VAF: 0.1%), but they are not suitable for contemporaneous targeting of multiple genomic regions [40].

The Idylla™ system is a fully automated liquid biopsy assay, which is currently available in Italy only for research purposes, as it is not approved for clinical use. The main advantage of Idylla™ relative to Therascreen^®^ or other CE-IVD approved assays is the short turnaround time because the cartridge-based testing solution tests the samples directly without the need for DNA extraction, which makes it particularly useful in clinical situations where a rapid genetic testing is necessary. In monitoring molecular changes, several studies have compared distinct methodologies, including targeted PCR-based assays, DNA microarray, NanoString technologies, and NGS approaches, evidencing different rates of mutation false results [41,42].

Our study showed extremely high agreement rates in detecting the most common *EGFR* alterations in lung adenocarcinomas (for detection of the T790M mutation, a 100% concordance rate between Idylla™ and Therascreen^®^ was observed). A similar result was reported by Heeke, who compared results of the Idylla™ system to those of an FDA-approved PCR-based assay among 54 NSCLC patients [41]. The authors found that detection of *EGFR* mutations was comparable but slightly lower with the Idylla™ device for *EGFR* exon 21 L858R mutations (14/15 detected, 93%) and exon 19 deletions (18/20 detected, 90%); concordance between the two methods in detecting T790M was 100%, exactly as in our study.

NGS technologies marked a revolution in the field of molecular testing, as they provide results of simultaneous sequencing of millions of DNA fragments, which may contain a high number of different mutations involving multiple genes from the same or different types of cancers. NGS can nowadays be carried out with limited quantities of DNA and provides information on a broader number of genetic alterations with high specificity and sensitivity, allowing a wider view of the mutational landscape of the disease [43]. The detection limit of the several NGS platforms commercially available is approximately 1–2% [40,44], and this may be one of the reasons that determined the low concordance between Therascreen^®^ and NGS assays in detecting EGFR mutations—mainly, for the T790M variant—in our study.

Other limitations of the NGS approaches are their high turnaround time and the elevated costs for testing in daily practice, along with some technical issues (such as the excessive DNA fragmentation) that impair the success of the variants’ detection through the NGS-based procedures. As an additional drawback, traditional short-read NGS platforms contemplate the use of PCR-amplified DNA libraries before sequencing, and this makes it impossible to definitively know whether two identical sequence reads originated from copies of the same starting molecule or from two independent molecules [40]. This issue has been solved with the introduction of unique molecular identifiers (UMI) or molecular barcodes in modern NGS platforms [45,46], though the modernization of all NGS systems involved in molecular diagnosis does not take place in a short time. In contrast to our findings, a recent study confirmed high reproducibility across PCR-based and NGS-based platforms in detecting EGFR mutation in plasma [47]. This strongly suggests that maximal attention be paid to all pre-analytic variables, which can influence the accuracy and reliability of NGS results. Efforts are required to standardize pre-analytical and analytical procedures according to the used testing methods.

Our study has some limitations, mainly the low number of cases included. However, the main purpose of our investigations was to compare both the accuracy and reliability of three methods already in use in clinical practice for further clarification on the methodology to be routinely applied in molecular diagnosis based on ctDNA specimens. Nevertheless, the triplicate screening assays carried out on the same 54 ctDNA samples as well as on the 48 genomic DNA samples from FFPE tumour tissues were pretty consistent. At the same time, our study is the first to compare a PCR-based CE-IVD-approved assay with Idylla™ and an NGS technique in liquid biopsy samples. Considering the 28 blood samples positive for at least one EGFR mutation, the combination of two of our methods allowed us to achieve an identical and quite complete coverage of variant detection: both real-time PCR assays or Idylla™ and NGS or Therascreen^®^ and NGS approaches identified the same very high rate of circulating mutations (27/28; 96.4%) (see Appendix A). Therefore, our findings strongly suggest considering a combination of mutation detection assays as the molecular screening approach of choice for routine assessment of EGFR mutational status based on ctDNA in NSCLC patients aimed at providing guidance toward the appropriate treatment strategy. In other words, the use of two sensitive molecular methods may ensure the highest level of diagnostic accuracy and minimize the risk of false-negative cases.

Recently, an implementation and technical validation of additional screening strategies for EGFR mutational status—such as the droplet digital PCR (ddPCR) assays or, more in general, the ultrasensitive PCR approaches as well as the NGS-based analytical assessment of the allele frequency of gene variants (VAF) [48,49,50]—using liquid biopsies or minimal cytological specimens from NSCLC patients may be helpful to identify a higher number of mutation-positive patients, providing predictive biomarkers of clinical behaviour.

Finally, in some clinical cases without tissue samples available for histological evaluation, the liquid biopsy may progressively represent a reliable tool for molecular testing, as also confirmed in our recent experience with an NSCLC patient who successfully underwent anti-EGFR-targeted therapy prescribed after a cfDNA screening [51].

## 4. Materials and Methods

### 4.1. Patients and Samples

Fifty-four liquid biopsy samples from 48 consecutive patients with advanced lung adenocarcinoma, who were treated with first- (Gefitinib; N = 41) and second- (Afatinib; N = 7) generation EGFR TKIs and needed T790M detection for treatment decision after disease progression, were enrolled into the study. A total of 54 ctDNA specimens were analysed because of multiple T790M determinations in some patients during treatment after progression. All patients were treated in the medical oncology units of North Sardinia, which refer to the National Research Council (CNR) of Sassari for molecular testing. Treatments and molecular tests were prescribed in all cases in accordance with the current guidelines of the Italian Association for Medical Oncology (AIOM). All patients were informed on the aims and methodology details of the study and gave their signed consent. The study was carried out in accordance with the principles of the Declaration of Helsinki and was approved by the Committee for the Ethics of the Research and Bioethics of the National Research Council/CNR n. 12629.

### 4.2. Mutation Testing

#### 4.2.1. Tissue DNA Isolation and Screening

Genomic DNA was isolated from biopsy tissue sections using a standard protocol. In particular, paraffin was removed from formalin-fixed paraffin-embedded (FFPE) samples with Bio-Clear (Bio-optica, Milan, Italy), and DNA was purified using the QIAamp DNA FFPE Tissue kit (Qiagen, Valencia, CA, USA). DNA quantitation and quality assessment were carried out with both a Nanodrop 2000 spectrophotometer (Thermo Scientific, Wilmington, DE, USA) and Qubit^®^ 2.0 Fluorometer (Invitrogen, Carlsbad, CA, USA). DNA fragmentation status was evaluated with an Agilent 2200 TapeStation system using Genomic DNA ScreenTape assay (Agilent Technologies, Santa Clara, CA, USA) able to produce a DNA integrity number (DIN). Quantitative measurements of *EGFR* mutations were performed using the Therascreen™ EGFR Pyro Kit (Qiagen), for the detection of mutations in codon 719 (exon 18), in codons 768 and 790 (exon 20), and in codons 858 to 861 (exon 21), along with deletions and complex mutations in exon 19 of the human *EGFR* gene on genomic DNA. Each pyrosequencing assay was performed on the PyroMark Q24 system (Qiagen), following the manufacturer’s instructions.

#### 4.2.2. ctDNA Isolation from Plasma

The flow chart of the workflow adopted for ctDNA extraction and for the tests performed is depicted in Figure 2.

A sample of about 8 mL of fresh peripheral blood collected in EDTA-containing tubes was first centrifuged at 1900× *g* for 10 min to plasma separation, then further centrifuged at 16,000× *g* for 10 min to remove any debris. About 4 mL of plasma was collected per patient; an aliquot of 2 mL was used to isolate ctDNA for the Therascreen^®^ and NGS approaches, and the remaining aliquot of 2 mL was processed on the Idylla™ System. The ctDNA was extracted using the QIAamp circulating nucleic acid kit on the QIAVac 24 Plus connected to a vacuum pump, according to the manufacturer’s instructions. The concentration of purified ctDNA was assessed using the Qubit 2.0 Fluorometer and the Qubit dsDNA HS (high sensitivity) assay kit (Life Technologies, Carlsbad, CA, USA). The cfDNA concentration for the analysis with NGS AmpliSeq Cancer Hotspot panel and Therascreen^®^ EGFR Plasma RGQ-PCR Kit as determined using the Qubit fluorimeter is typically between 10 and 50 nanograms, respectively.

#### 4.2.3. ctDNA Screening: Therascreen^®^

The Therascreen^®^ EGFR Plasma RGQ PCR kit was used as a reference method and performed according to the manufacturer’s instructions; this involves a real-time PCR assay that combines an amplification refractory mutation system (ARMS) and a Scorpion fluorescent primer/probe system. Allele-specific amplification is achieved by the ARMS assay, which exploits the ability of Taq DNA polymerase to distinguish between a matched and mismatched base at the 3′ end of a PCR primer. When the primer is fully matched, the amplification proceeds with full efficiency. When the 3′ base is mismatched, only low-level background amplification may occur. Therefore, a mutated sequence is selectively amplified even in samples where most of the DNA does not carry the mutation. Detection of amplification is performed using Scorpions. Scorpions are bifunctional molecules containing a PCR primer covalently linked to a probe. The probe incorporates the fluorophore carboxyfluorescein (FAM™) and a quencher. The latter quenches the fluorescence of the fluorophore. When the probe binds to the ARMS amplicon during PCR, the fluorophore and quencher become separated, leading to a detectable increase in fluorescence. The Therascreen^®^ EGFR RGQ PCR kit enables the detection of 21 mutations: 19 deletions in exon 19, T790M in exon 20 and L858R missense mutations in exon 21. A Rotor-Gene Q MDx instrument was used to perform the real-time qualitative PCR assay for the detection of somatic mutations in the *EGFR* gene, using genomic DNA extracted from liquid biopsy samples. A list of *EGFR* mutations detected by the Therascreen^®^ assay is reported in Appendix A.

#### 4.2.4. ctDNA Screening: Idylla™

The second aliquot of 2 mL of plasma of each patient was processed on the Biocartis Idylla™ System, a fully automated liquid biopsy assay. The Idylla technology is cartridge-based and uses microfluidic (capillary action-based pumping) processing with all reagents on-board, particularly tested in colorectal cancer specimens from formalin-fixed, paraffin-embedded tissue sections [52]. The cartridges are loaded with 2 ml of plasma and 20 µL of Proteinase K (20 mg/mL), and the remaining processes, including extraction and real-time PCR of EGFR ctDNA, data analysis and results, are completely automated. The Idylla technology combines allele-specific primer amplification using PlexPrimers with allele-specific detection using PlexZymes (also known as MNAzymes) [53,54]. Each PlexPrimer contains an “insert sequence”, positioned between 5′ and 3′ target-specific regions, which is non-complementary to the target initially, but is introduced into amplicons during amplification. For multiplexed mutation detection, each PlexPrimer, containing a different INS sequence, is designed to be allele-specific via complementarity of the 3′ terminus of the target mutation [54]. The Idylla™ ctEGFR Mutation Assay is designed for research purposes and covers 40 different *EGFR* variations: G719A/C/S in exon 18; T790M and S768I in exon 20; L858R and L861Q in exon 21; 28 deletions in exon 19 and 5 insertions in exon 20. A list of *EGFR* mutations detected by the Idylla™ assay is reported in Appendix A.

#### 4.2.5. ctDNA Screening: Next-Generation Sequencing (NGS)

NGS experiments were carried out using the Ion Torrent S5 System. The ready-to-use Ion AmpliSeq Cancer Hotspot Panel v.2 (CHPv2 250 amplicons) (ThermoFisher Scientific, Waltham, MA, USA) was used to evaluate the mutational status of approximately 2800 COSMIC (catalogue of somatic mutation in cancer) mutations from 50 oncogenes and tumour suppressor genes. Libraries were generated starting from 10 ng of input ctDNA with the Ion AmpliSeq Library Kit Plus2.0 and barcoded with Ion Xpress Barcode Adapters (Life Technologies). After dilution to a final concentration of 50 pM, libraries were pooled together, then placed in the Ion Chef for emulsion PCR and Chip (520−530) loading steps. Sequencing of libraries loaded on the Chip was performed with the Ion S5^TM^ System (Thermofisher) using the recommended reagents. Sequencing data were processed with the Ion Torrent platform-specific pipeline software (Torrent Suite, V5.2.1). Ion Reporter™ V5.12 and Integrative Genome Viewer software (http://www.broadinstitute.org/igv accessed on 11 March 2023) were used for variant annotation and read visualizations, respectively (Appendix A).

#### 4.2.6. Limits of Detection of the Three Mutation Assays

The Idylla™ ctEGFR mutation assay (Biocartis) has an estimated limit of detection (LOD) of 0.2% of allele frequency. This means that the Idylla™ ctEGFR system can detect mutations in a sample at an allele frequency of 0.2% or higher. The LOD for the Therascreen^®^ EGFR Plasma RGQ-PCR Kit (Qiagen) ranges between 0.15 and 0.2% allele frequency. The LOD for the Ion AmpliSeq Cancer Hotspot panel (ThermoFisher) is determined by the grade of detection sensitivity into the targeted sequence and the sequencing depth. With such an NGS-based system, commercial targeted panels can reliably have a limit of detection down to 0.01%, with a sequencing depth depending on the quality of the DNA isolated from the biological specimen. The LOD is typically lower in deeper sequencing runs, which provide more coverage of the targeted regions. The Ion AmpliSeq Cancer Hotspot panel has been designed to detect and accurately genotype most hotspot variants associated with many common cancer types, including lung adenocarcinoma. In addition to the sensitivity for the targeted sequences, the accuracy of the LOD for the Ion AmpliSeq Cancer Hotspot panel is also dependent on the quality of the sample. Poor-quality samples can reduce the accuracy of the LOD and may require additional sequencing depth or additional strategies to obtain an accurate result.

### 4.3. Statistical Analysis

A descriptive analysis of categorical variables using percentages was performed, and differences between them were evaluated by Fisher test or chi-squared test, as appropriate. Continuous data were expressed as medians and ranges. Analyses were performed with MedCalc Windows, version 19.4.1 (MedCalc Software, Ostend, Belgium).

## 5. Conclusions

The availability of multiple screening techniques—the Therascreen^®^ test officially approved for clinical practice, the fully automatized Idylla™ system, and the NGS targeted panels—allow us to achieve high diagnostic performance in detecting EGFR. Each single tumour specimen should be addressed by a different diagnostic assay according to its pathological or biological features; as a consequence of this, a laboratory for molecular diagnosis in oncology should possess more than one instrumental platform or system in order to offer the possibility of having access to the most appropriate diagnostic tool.

## Figures and Tables

**Figure 1 ijms-24-06410-f001:**
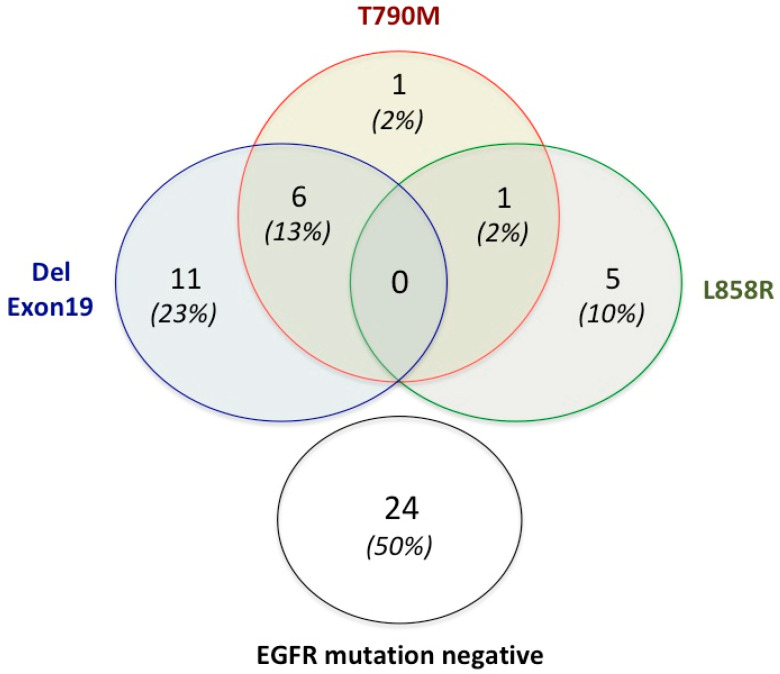
EGFR mutation status in ctDNA samples at the time of disease progression using the Therascreen^®^ assay. Number and percentages of the different subsets of patients (N = 48) are reported.

**Figure 2 ijms-24-06410-f002:**
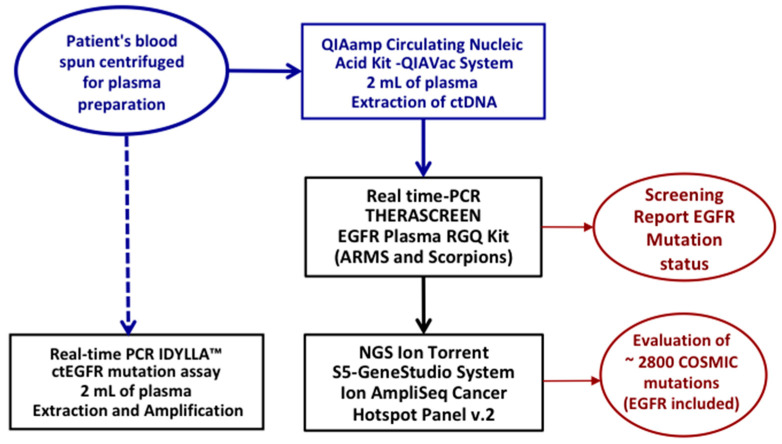
Flow chart of the molecular screening process of the samples included in the study.

**Table 1 ijms-24-06410-t001:** Demographic and clinical characteristics of the patients enrolled in the study.

Characteristics	Total of Patients (%)
** *Age at diagnosis, years* **	Median 70 (range 41–85)
** *Gender* **	
Male Female	13 (27.1) 35 (72.9)
** *Smoking status* **	
Never Smokers Former Smokers Smokers	34 (70.8) 5 (10.4) 9 (18.8)
** *Disease Stage* **	
IIIB IV	20 (41.7) 28 (58.3)
** *Disease sites* **	
Single location Multiple locations * With CNS involvement* * Without CNS involvement*	29 (60.4) 19 (39.6) *4* (*8.3*) *15* (*31.3*)
** *Type of treatment* **	
First-generation EGFR TKI (Gefitinib) Second-generation EGFR TKI (Afatinib)	41 (85.4) 7 (14.6)

CNS: central nervous system; EGFR: epidermal growth factor receptor; TKI: tyrosine kinase inhibitor.

**Table 2 ijms-24-06410-t002:** *EGFR* mutation status comparison between testing on primary tumour tissue with pyrosequencing and on blood ctDNA at progression with Therascreen.

EGFR Mutation	FFPE (n. %)	ctDNA+ (n. %)	ctDNA− (n. %)
*Exon 18 G719A/S*	3 (6.2%)	ND	ND
*Exon 19 Deletions*	31 (64.6%)	20 (64.5%)	11 (35.5%)
*Exon 21 L858R*	12 (25.0%)	7 (58.3%)	5 (41.7%)
*Exon 21 L861Q*	2 (4.2%)	ND	ND

ctDNA: circulating tumour DNA; FFPE: formalin fixed paraffin embedded. ND: not determined, since not included in the Therascreen assay.

**Table 3 ijms-24-06410-t003:** *EGFR* mutation status on primary tumour tissue and blood ctDNA at progression according to the disease features.

		Exon 19 Deletions			L858R		
Metastasis	Patients	FFPE	ctDNA+	*p*	FFPE	ctDNA+	*p*
1 site	29 (60.4%)	18 (62)	10/18 (56)	0.402	8 (28)	4/8 (50)	0.394
≥2 sites	19 (39.6%)	13 (68)	10/13 (77)	0.418	4 (21)	3/4 (75)	0.270
**Stage**	**Patients**	**FFPE**	**ctDNA+**		**FFPE**	**ctDNA+**	
IIIB	20 (41.7%)	11 (55)	7/11 (64)	0.426	8 (40)	4/8 (50)	0.690
IV	28 (58.3%)	20 (71)	13/20 (65)	0.385	4 (14)	3/4 (75)	0.150

ctDNA: circulating tumour DNA; FFPE: formalin fixed paraffin embedded. Statistical significance (*p*) at 0.05.

**Table 4 ijms-24-06410-t004:** Concordance rates between methods used to screen the 54 ctDNA samples.

Sample	Thera Screen	Idylla	% Agreement Idylla with Therascreen	*p*	NGS	% Agreement NGS with Therascreen	*p*
Total ctDNA	54	49	90.7	0.057	40	74.1	<0.001
ctDNA+	27	23	85.2	0.111	13	48.1	<0.001
ctDNA−	27	26	96.3	1.000	27	100	1.000

ctDNA: circulating tumour DNA; NGS: next-generation sequencing. Statistical significance (*p*) at 0.05.

**Table 5 ijms-24-06410-t005:** Comparisons between Therascreen^®^, Idylla™, and NGS in detecting the most clinically relevant EGFR gene alterations in ctDNA samples.

EGFR Alteration	Therascreen	Idylla	Agreement %	*p*
T790M mutation	8/54	8/54	100	1.000
Exon19 deletion	27/54	23/54	85.2	0.562
L858R mutation	17/54	18/54	94.4	1.000
**EGFR alteration**	**Therascreen**	**NGS**		
T790M mutation	8/54	3/54	37.5	0.201
Exon19 deletion	27/54	20/54	74.1	0.244
L858R mutation	17/54	15/54	88.2	0.833

EGFR: Epidermal Growth Factor Receptor; NGS: next-generation sequencing. Statistical significance (*p*) at 0.05.

## Data Availability

All data generated or analysed during this study are included in this published article and its Supplementary Information files.

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
