# Peer review of "Comparison between Three Different Techniques for the Detection of EGFR Mutations in Liquid Biopsies of Patients with Advanced Stage Lung Adenocarcinoma"

_ijms, 2023, doi:10.3390/ijms24076410_

Round 1

Reviewer 1 Report

Title: Comparison between three different techniques for the detection of EGFR mutations in liquid biopsies of patients with advanced stage lung adenocarcinoma

Summary: Authors compared three different techniques for EGFR detection in liquid biopsies of late stage lung adenocarcinoma patients. Overall the study looks well designed. However, I have following comments that authors should address:

Major comments:

1.       Fig 1: Authors should also add number of samples for each section in venn diagram.

Minor comments:

1.       Line 27 in Abstract: “for EGFR detection in liquid biopsies of such patients” should be changed to “for EGFR mutation detection in liquid biopsies of such patients”.

Author Response

Summary: Authors compared three different techniques for EGFR detection in liquid biopsies of late stage lung adenocarcinoma patients. Overall the study looks well designed. However, I have following comments that authors should address:

Major comments:

Fig 1: Authors should also add number of samples for each section in venn diagram.

REPLY: Fully agreeing with the suggestion of the Reviewer, we completely modify the Figure 1 in order to provide number and percentages of the different subsets of cases. We accordingly modified the Figure legend. Into the text (paragraph 1 at page 4 of this revised version), we have specified as follows: “Figure 1 depicts the results of testing with Therascreen® in blood samples from the series at the time of the disease progression, reporting the subtypes of EGFR mutations detected by this assay”. At the same level (last paragraph at page 3 and beginning of paragraph 1 at page 4), we have clarified that: “For some patients, more than one ctDNA specimen was analysed due to the clinical needs - i.e. sequential disease progressions - requiring multiple T790M determinations before treatment decision. Overall, 54 ctDNA samples from 48 patients underwent mutation screening.”

Reviewer 2 Report

In this manuscript, the authors compare the diagnostic value of liquid genetic testing with that of tissue testing. In existing clinical applications, especially for patients with advanced lung cancer, we are often faced with the problem of difficulty in obtaining tissue to allow histological and genetic testing. The advent of liquid biopsy is a complement to the methodology. However, some scholars have suggested that the accuracy of liquid biopsy is much less new than tissue testing. This manuscript demonstrates just how comparable the diagnostic accuracy of the two is. Thus, this manuscript provides supporting evidence for the use of liquid biopsy in patients with advanced lung cancer.

This manuscript has some problems, such as too small a sample size. If the authors had been able to collect more patients to illustrate this issue, the conclusions would have been more reliable.

Author Response

In this manuscript, the authors compare the diagnostic value of liquid genetic testing with that of tissue testing. In existing clinical applications, especially for patients with advanced lung cancer, we are often faced with the problem of difficulty in obtaining tissue to allow histological and genetic testing. The advent of liquid biopsy is a complement to the methodology. However, some scholars have suggested that the accuracy of liquid biopsy is much less new than tissue testing. This manuscript demonstrates just how comparable the diagnostic accuracy of the two is. Thus, this manuscript provides supporting evidence for the use of liquid biopsy in patients with advanced lung cancer.

This manuscript has some problems, such as too small a sample size. If the authors had been able to collect more patients to illustrate this issue, the conclusions would have been more reliable.

REPLY: Although we agree with the Reviewer on this aspect, we have better specified in last paragraph of the Discussion section the aims of the study by adding the following sentences: “Our study has some limitations, mainly the low number of cases included. However, the main purpose of our investigations was to compare both accuracy and reliability of three methods already in use in clinical practice for further clarification on the methodology to be routinely applied into molecular diagnosis on ctDNA specimens. Nevertheless, considering the triplicate screening assays carried out on the same 54 ctDNA samples as well as on the 48 genomic DNA samples from FFPE tumour tissues, the collection of analysed specimens is pretty consistent.

Reviewer 3 Report

Authors need to state the limit of detection used for all platforms.  Also need evidence of this at least in supplementary data e.g. serial dilutions with oligonucleotides, particularly for the new Idylla platform.  

Supplementary Table 1 needs editing to remove all symbols.

Do not feel that Supplementary Table 2 adds anything to paper

Author Response

Authors need to state the limit of detection used for all platforms. Also need evidence of this at least in supplementary data e.g. serial dilutions with oligonucleotides, particularly for the new Idylla platform.

REPLY: To provide full details on detection limits of the screening assays used in our study, in Methods (at page 10 of the revised manuscript) the following new paragraph has been added:

4.2.6 Limits of detection of the three mutation assays

The Idylla™ ctEGFR mutation assay (Biocartis) has an estimated limit of detection (LOD) is 0.2% of allele frequency. This means that the Idylla™ ctEGFR system can detect mutations in a sample at an allele frequency of 0.2% or higher. The LOD for the Therascreen® EGFR Plasma RGQ-PCR Kit (Qiagen) is ranging between 0.15 and 0.2% allele frequency. The LOD for the Ion AmpliSeq Cancer Hotspot panel (ThermoFisher) is determined by the grade of detection sensitivity into the targeted sequence and the sequencing depth. With such NGS-based system, the LOD can range from 0.01% to 5%, depending on the sample type and sequencing depth. The LOD is typically lower in deeper sequencing runs, which provide more coverage of the targeted regions. The Ion AmpliSeq Cancer Hotspot panel has been designed to detect and accurately genotype most hotspot variants associated with many common cancer types, including lung adenocarcinoma. In addition to the sensitivity of the targeted sequences, the accuracy of the LOD for the Ion AmpliSeq Cancer Hotspot panel is also dependent on the quality of the sample. Poor-quality samples can reduce the accuracy of the LOD and may require additional sequencing depth or additional strategies to obtain an accurate result.”

Moreover, lists of EGFR mutations detected by Therascreen® and Idylla™ assays have been reported in Supplementary Tables 2 and 3, respectively. The list of EGFR mutations detected by Ion AmpliSeq Cancer Hotspot panel became the Supplementary Table 4 in this revised version of the manuscript.

Supplementary Table 1 needs editing to remove all symbols. Do not feel that Supplementary Table 2 adds anything to paper

REPLY: Symbols in Supplementary Table 1 have been removed. Supplementary Table 2 of the previous version of the manuscript has been deleted and the text modified accordingly.

Round 2

Reviewer 1 Report

Authors have addressed the comments. The study is acceptable in the current form.

Author Response

We would like to thank Reviewer 1 to consider addressed our comments and retain the study acceptable in the current form.

Reviewer 3 Report

See below 

Author Response

Unfortunately, we did not find any attachment at the section "Comments and Suggestions for Authors
" (though there is an indication "See below")